Unexpected mitochondrial lineage diversity within the genus Alonella Sars, 1862 (Crustacea: Cladocera) across the Northern Hemisphere

Neretina Anna N. 1
Karabanov Dmitry P. 1 2
Sacherova Veronika 3
Kotov Alexey A. alexey-a-kotov@yandex.ru 1
1 A.N. Severtsov Institute of Ecology and Evolution, Russian Academy of Sciences , Moscow , Russia
2 I.D. Papanin Institute for Biology of Inland Waters , Borok , Yaroslavl State , Russia
3 Charles University Prague , Prague , Czech Republic
Poyarkov Nikolay
Electronic publication date: 2021 Feb 1
Publication date: 2021
Volume: 9
Electronic Location ID: e10804
Received 2020 Sep 11; Accepted 2020 Dec 30
Copyright: ©2021 Neretina et al.
Copyright year: 2021
Copyright holder: Neretina et al.
License: This is an open access article distributed under the terms of the Creative Commons Attribution License, which permits unrestricted use, distribution, reproduction and adaptation in any medium and for any purpose provided that it is properly attributed. For attribution, the original author(s), title, publication source (PeerJ) and either DOI or URL of the article must be cited.
License URL: https://creativecommons.org/licenses/by/4.0/

Keywords: Cladocera, Phylogeography, Genetics, Biodiversity, Biogeography

Funding: Russian Science Foundation 18-14-00325 This work was supported by Russian Science Foundation (grant 18-14-00325). The funders had no role in study design, data collection and analysis, decision to publish, or preparation of the manuscript.

==============================
Representatives of the genus Alonella Sars (Crustacea: Cladocera: Chydorinae) belong to the smallest known water fleas. Although species of Alonella are widely distributed and often abundant in acidic and mountain water bodies, their diversity is poorly studied. Morphological and genetic approaches have been complicated by the minute size of these microcrustaceans. As a result, taxonomists have avoided revising these species. Here, we present genetic data on Alonella species diversity across the Northern Hemisphere with particular attention to the A. excisa species complex. We analyzed 82 16S rRNA sequences (all newly obtained), and 78 COI sequences (39 were newly obtained). The results revealed at least twelve divergent phylogenetic lineages, possible cryptic species, of Alonella, with different distribution patterns. As expected, the potential species diversity of this genus is significantly higher than traditionally accepted. The A. excisa complex is represented by nine divergent clades in the Northern Hemisphere, some of them have relatively broad distribution ranges and others are more locally distributed. Our results provide a genetic background for subsequent morphological analyses, formal descriptions of Alonella species and detailed phylogeographical studies.

Introduction

Water fleas (Crustacea: Cladocera) are microscopic crustaceans common to continental waters (Kotov, 2013). Daphnia O.F. Müller is familiar to the public as a study subject in classrooms and as a food source in the aquarium industry. But related cladocerans that are crucial elements in littoral and benthic ecosystems are mostly unknown to the public. A rough estimate of the approximate number of cladoceran individuals in the World, based on the total area of inland waters being 106 km2 and an underestimated average number of cladocerans being 1,000 individuals per 1 m2, gives 1015 individuals (Smirnov & Kotov, 2010). Most of these cladocerans do not belong to the genus Daphnia, but their ecological importance is immense and very little is known about their diversity.

While Daphnia is universally accepted as an important model for ecological, toxicological and genetic studies (Lampert, 2011), we still know very little about other water fleas. However, in the last ten years, substantial progress has been made by integrative taxonomic and phylogenetic studies of non-model species groups from the families Daphniidae (Ishida, Kotov & Taylor, 2006; Petrusek et al., 2008; Dlouhá et al., 2010; Kotov & Taylor, 2010; Popova et al., 2016; Bekker et al., 2018; Karabanov et al., 2018; Kotov & Taylor, 2019; Kotov et al., 2020), Bosminidae (Kotov, Ishida & Taylor, 2009; Faustova et al., 2011; Faustova, 2017), Eurycercidae (Bekker, Kotov & Taylor, 2012), Moinidae (Petrusek, Černy & Audenaert, 2004; Bekker et al., 2016; Montoliu-Elena, Elías-Gutiérrez & Silva-Briano, 2019; Ni et al., 2019), Chydoridae (Sacherová & Hebert, 2003; Belyaeva & Taylor, 2009; Kotov et al., 2016; Sinev, Karabanov & Kotov, 2020), Polyphemidae (Xu et al., 2009) and Leptodoridae (Xu et al., 2011). Based on these works, water fleas are no longer considered as exemplars for cosmopolitanism (Frey 1982; Frey 1987). There has also been significant progress in large-scale biogeographic reconstructions for these tiny animals. But the slogan “everything is everywhere” (Baas Becking, 1934) still may be applied to the biogeography of taxonomically difficult groups of cladocerans, such as the genus Alonella Sars (Anomopoda: Chydoridae: Chydorinae). This genus includes the smallest representatives of the water fleas known to date. The adult specimens of Alonella do not exceed 0.45 mm in length and are barely visible to the naked eye (Smirnov, 1971; Smirnov, 1996). Although Alonella is widely distributed all around the World and often abundant in acidic and mountain water bodies (Smirnov, 1996; Van Damme & Eggermont, 2011), its diversity is still poorly studied. Morphological and genetic study of Alonella is made difficult by the small body size. Although some other small-bodied cladoceran taxa are intensively studied now, taxonomists have avoided revision of the species of Alonella. Since N.N. Smirnov’s monographs on the chydorids (Smirnov, 1971; Smirnov, 1996), only a single new species of Alonella has been described (Alonso & Kotov, 2017). Also, few attempts to isolate and sequence DNA have been carried out for this genus mainly due to the molecular barcoding efforts (Costa et al., 2007; Jeffery, Elías-Gutiérrez & Adamowicz, 2011; Prosser, Martínez-Arce & Elías-Gutiérrez, 2013), and studies of the chydorid generic relationships (Sacherová & Hebert, 2003).

In fact, only three morphospecies of Alonella: A. nana (Baird), A. exigua (Lilljeborg) and A. excisa (Fischer) (Figs. 1A–1F) are recognizable now in the Northern Hemisphere following the Smirnov’s key (Smirnov, 1996). A. nana is characterized by a sub-globular body shape and prominent diagonal lines on the valves (Smirnov, 1971; Hudec, 2010) (Figs. 1A–1B). A. excisa and A. exigua have an elongated body and polygonal ornamentation (Figs. 1C–1F). There are small dots within each polygon in A. exigua (Figs. 1C–1D), while each polygon in A. excisa carries short parallel striation (Figs. 1E–1F). Each of these three morphospecies has a very wide geographic range. As such, they are candidates for groups of sibling species (Frey, 1982; Frey, 1987). Indeed, preliminary morphological evidence suggested the existence of several species within the A. excisa complex (Kotov et al., 2013). The status of other Alonella and Alonella-like taxa (out of nana, excisa and exigua groups) (Smirnov, 1971; Smirnov, 1996) remains unclear. The aim of the present work is to elucidate the mitochondrial lineage diversity and preliminary biogeography of Alonella (especially of the A. excisa species complex) across the Northern Hemisphere using different methods of the OUT delimitation. The evidence is based on variation in mitochondrial 16S ribosomal RNA (16S) and cytochrome c oxidase subunit I (COI) genes.

Figure 1 Alonella parthenogenetic females identified based on morphological characters.

General view and sculpture of valves of A. nana (A–B) (from Lake Glubokoe, Moscow Area, Russia); A. exigua (C–D) (from Meertvoe Lake in the vicinities of Krasnaya Pahra village, Moscow Area, Russia); A. excisa (E–F) (from the roadside mire, Yakutia Republic, Russia). Scale bars: 0.1 mm for A, C, E; 0.02 mm for B, D, F.

Material and Methods

Ethics statement

Field collection in public property in Russia does not require permissions. Sampling in the state natural reserves of Russia was conducted with special verbal permission of their heads (O.P. Elizarova, Pinezhsky State Natural Reserve; T.I. Shpilenok, Kronotsky Biosphere Reserve; Y.P. Sushitsky, Khanka Nature Reserve). Ethiopian samples were collected in the frame of the Joint Ethio-Russian Biological Expedition, with permission of Ministry of Environment, Forest and Climate Change of Ethiopia. Samples in Mongolia were taken in the frame of the Joint Russian-Mongolian Complex Biological Expedition with special permission of the Ministry of Nature, Environment and Tourism of Mongolia. Samples in South Korea were collected in the frame of cooperation between A.A. Kotov and the National Institute of Biological Resources of Korea and does not require special permission. All the localities in Ethiopia, Mongolia and South Korea belong to public property. They are relatively large lakes, streams, affluents of rivers or roadside ponds.

Field collection, identification and photographing

Samples were collected by plankton nets (with mesh size of 30–50 µm) of different construction and fixed immediately after sampling in 96% ethanol. All samples were preliminarily inspected in the laboratory under a binocular stereoscopic microscope LOMO (Open Joint-Stock Company, Russia). In samples where Alonella taxa were detected, the whole volume of sample was examined under light microscope Olympus BX41 for accurate identification based on morphological characters via standard keys (Smirnov, 1971; Smirnov, 1996). Selected individuals were studied under a scanning electron microscope CamScan MV 2300 (Tescan, Czech Republic) as described previously (Kotov, 2013; Neretina & Kotov, 2015).

DNA sequencing

One to three parthenogenetic females from each population (see Table S1) predominantly of the A. excisa species complex were studied for genetic profiles. Identification of each parthenogenetic female used in the genetic analysis was especially re-checked under a stereoscopic microscope in order to avoid possible mistakes related with situations when a sample contained several Alonella species simultaneously. Selected individuals were placed into 96-well PCR plates and air-dried. DNA from individual crustaceans was extracted in 50 µl of proteinase K solution, according to the protocol of Schwenk et al. (1998)). PCR reactions were carried out in 25 µL volume, containing 5 µL of genomic DNA, 6.2 µL of distilled H2O, 0.65 µL (10 µM) of each primer to amplify either COI (COI-F: 5′-TGTAAAACGACGGCCAGTTCTASWAATCATAARGATATTGG-3′; COI -R: 5′-CAGGAAACAGCTATGACTTCAGGRTGRCCRAARAATCA- 3′) or 16S (16S-F: 5′-CCGGAATTCCGCCTGTTTATCAAAAACA-3′; 16S-R: 5′-CCCAAGCTTCTCCGGTTTGAACTCAGAT-3′) (see details on COI primers in Prosser, Martínez-Arce & Elías-Gutiérrez (2013) and details on 16S primers in Simon et al. (1994)) and 12.5 µL of PPP Master Mix (Top-Bio, the Czech Republic) in a thermocycler GeneTouch (Hangzhou Bioer Technology Co., China). The PCR cycles both for COI and 16S fragments included the following steps: initial denaturation at 92 °C for 3 min, 40 cycles (denaturation at 94 °C for 1 min, annealing at 50 °C for 1 min, and elongation at 72 °C for 1.5 min), and final elongation at 72 °C for 5 min. Amplified PCR products were sequenced using forward and reversed primers or only via forward primers. In the first case, a single consensus sequence was assembled using the forward and reverse sequences using CodonCode Aligner v. 6.0.2 (CodonCode Corp, USA) and checked for possible stop-codon presence. DNA sequences were submitted to the NCBI GenBank database (accession numbers MN608113 –MN608151 for COI and MN598677 –MN598759 for 16S) (Table S1).

Analysis of sequences and reconstruction of phylogeny

The authenticity of all newly obtained sequences was verified by BLAST comparisons (Boratyn et al., 2013). We also added two species of Chydoridae (Alona affinis (Leydig, 1860) and A. setulosa Megard, 1967) as outgroups and existing sequences of Alonella from GenBank (Table S1) to our study. The sequences were aligned via a software package MAFFT v.7 (Katoh & Standley, 2016) on the server of the Computational Biology Research Center, Japan (http://mafft.cbrc.jp). For alignment of the protein coding COI locus we used the “Translation Align” option with strategy FFT-NS-i. For the 16S locus, we used the Q-INS-i algorithm, which considers secondary structure. Searching of appropriate models and partitioning schemes were fulfilled via ModelFinder v.1.6.9 (Kalyaanamoorthy et al., 2017) on the web-service of the Center for Integrative Bioinformatics Vienna, Austria (http://www.iqtree.org). Selection of the best model was carried out based on the best minimal values of the Bayesian information criterion (BIC) (Posada & Buckley, 2004). Parameters of nucleotide substitutions were identified in ModelFinder (Kalyaanamoorthy et al., 2017) as K3Pu+F+G4 for 16S, and for COI triplets: 1st –TN+F+I, 2nd –TVM+F+I and 3rd –HKY+F+G4. Parameters of the model BIC were almost identical to those found via the corrected Akaike’s information criterion (AICc) (Posada & Crandall, 2001).

Phylogeny reconstruction was carried out for each locus separately. Also, we reconstructed a joint consensus tree based on the whole set of unlinked data using the maximum likelihood (ML) or Bayesian (BI) methods. For ML analysis we used an algorithm IQ-TREE v.1.6.9 (Nguyen et al., 2015), as implemented on the CIBIV web-server (Trifinopoulos et al., 2016). Each set of sequences was analyzed based on the best model, which was automatically calculated by W-IQ-TREE (Trifinopoulos et al., 2016). As the branch supporting test, we used 1k replics in UFbootstrap2, requiring significantly smaller computational resources as compared with traditional supporting tests and demonstrating a higher effectiveness of such calculations (Hoang et al., 2018). When conducting BI, the posterior probabilities (Bolstad, 2007) were calculated in BEAST2 v.2.6 (Bouckaert et al., 2019). All parameters of substitution models were identified for BI-trees via BEAUti (Drummond et al., 2012) (part of BEAST2 package) using the BIC criterion. In each BEAST2-analysis, we conducted four independent runs of MCMC (100M generations, with selection of each 10k generation) with effectiveness control in “R We There Yet” (RWTY) for “R” statistical language (Warren, Geneva & Lanfear, 2017). A consensus tree based on the maximum clade credibility (MCC) was obtained in TreeAnnotator v.2.6 (Drummond et al., 2012) with half increased burn-in rate determined in RWTY (but no less than 20%).

ML-testing MEGA-X (Kumar et al., 2018) rejected a strict molecular clock. Therefore, we used an uncorrected relaxed clock with lognormal distribution, allowing to vary the substitution speed in different portions of the tree (Drummond et al., 2006). Speciation was analyzed using the Yule process approximation (Steel & McKenzie, 2001). Alona sequences from the GenBank are used as a priori designated outgroup. Having no additional information on stationary frequencies and parameters in nucleotides substitution, we ignored the priors of Dirichlet’s distribution due to their weak positive influence on the phylogeny reconstructions (Sarver et al., 2019). After conclusion on the full consensus in the main clades between BI and ML, we represented in our illustrations only ultra-metric BI trees, with branches supports for key nodes only.

Correlation between phylogenies based on different genetic loci is a special issue in every phylogenetic reconstruction (Nei & Kumar, 2000). We carried out a comparison between trees made in BEAST2 separately for 16S and COI, analyzing sequences exactly from the same vouchers on the tanglegram constructed in Dendroscope-3 (Scornavacca, Zickmann & Huson, 2011). General topologies of reconstructed 16S and COI trees were similar, which allowed us to analyze not only individual gene phylogenies, but also to use a more powerful coalescent methods to analyze the relationship between the reconstructed trees through the calculation of a multigenic supermatrix (BEAST2 (Bouckaert et al., 2019)) and by merging individual gene trees (ASTRAL-III (Zhang et al., 2018)). No fundamental differences of both tree topologies were found. We deleted branches with low support (Zhang et al., 2018). However, this transformation failed to improve the output tree. Thus, we have a justification for our reconstruction of multilocus phylogeny and combination of data even in the presence of “gaps” (Molloy & Warnow, 2018) as we did not have sequences of both genes from all specimens.

Species delimitation

There is no universal approach for species delimitation based on the OTUs in the gene sequence-based trees (Kartavtsev, 2018), and we used three most common approaches to the OTUs delimitation based on single locus data, as well as the whole dataset. Since a preliminary prior data sorting on the possible OTUs is required for most computer packages, we conducted an analysis of the tree reconstruction for each locus separately based on the ABGD distance method, coalescence models in the ‘splits’ and ‘bGMYC’ packets, as well as through the Poisson analysis of mPTP.

The simplest method based on analysis of a threshold of divergence, Automatic Barcode Gap Discovery (ABGD) (Puillandre et al., 2012), was realized based on the web-server Atelier de BioInformatique, France (http://wwwabi.snv.jussieu.fr/public/abgd/abgdweb.html). Single values for both mitochondrial loci were selected by us: Pmin = 0.001, Pmax = 0.1, Steps = 100, X = 10, Nb = 25.

The second method, applying the coalescence approach based on general mixed Yule-coalescent model (GMYC) (Pons et al., 2006) with the “classic” implementation of GMYC, was performed in the ‘splits’ package (Fujisawa & Barraclough, 2013) for Microsoft “R-Open & MKL” software v.3.5.3 x64 (http://mran.microsoft.com/). As the input tree, we used an ultrametric BI-tree made in BEAST2 for each locus. As it is known that realization of GMYC in the case of a complicated structure of natural populations leads to considerable over-estimation of the number of recognizable taxonomic units (Lohse, 2009), we used Bayesian realization of the general mixed Yale model and coalescence in order to increase (at least partly) the reliability of GMYC conclusions (Reid & Carstens, 2012) in the package ‘bGMYC’ for “R”. Input trees for ‘bGMYC’ were the same as for the ‘splits’ analysis. Sorting, re-rooting of the trees and removing the outgroups was carried out via the “R” package according to the script of Sweet et al. (2018). We randomly selected 100 ultrametric trees from the primary set to the ‘bGMYC’ processing with the following parameters: 100 k MCMC generations with 50% annealing; the range of threshold values from 2 to the maximum number of sequences in the data set; start values for both Yale and coalescence models according to Reid & Carstens (2012) as the most usable for the majority datasets. These sets allowed us to obtain the distribution of “Coalescence to Yule”>>0, what is a sign of a good fit of the model parameters to the data. The threshold level of the cladogenesis reliability was accepted as P > 0.95 and P > 0.99, which allows us to reduce the probability of an excessive lumping in the taxonomic structure.

The third method of the species delimitation was as the previous one, but based on the Poisson tree processes (PTP). This approach aims to distinguish speciation processes among the species from diversification processes within the species, but it analyzes the number of substitutions between branching events instead of time intervals. For data processing, we used multi-rate Poisson Tree Processes, mPTP (Kapli et al., 2017) on the web-server of Heidelberg Institute for Theoretical Studies (http://mptp.h-its.org/). As the input tree, we used the phylogenetic ML-tree obtained used W-IQ-TREE for each locus.

A new version of “tr2” (Fujisawa, Aswad & Barraclough, 2016) for Python v.3.7 x64 (http://www.python.org) was used as a method for multi-locus taxonomy. This method is based on the identification of a transition point between species branching and branching within species via estimation of observed and expected levels of genes congruence according to the coalescence theory for rooted triplets topologies. We used an option of testing of the OTUs apriori partitioning on the consensus ultrametric tree in BEAST2 for both loci unlinked. However, this mechanistic approach does not allow to adjust the model taking into account the genetic features and biology of different groups of organisms. As an alternative method, we used a Bayesian approach for the delimitation of multi-species coalescence model using molecular sequences from multiple loci in STACEY v.1.2.4 (Jones, 2017) for BEAST2. In fact, this method is a version of the multi-species coalescence model used in *BEAST (Heled & Drummond, 2009). There the birth-death-collapse model is used in order to estimate the species tree (Jones, 2017). Final phylogenetic relationships were estimated via STACEY in four independent runs for the whole data set. Each run consisted of 50M MCMC generations, with selection of every 10k with 10% pre-annealing. STACEY log files were examined in Tracer v.1.7.1 (Rambaut et al., 2018) to assess whether the runs have reached the stationary phase and converge on model parameters (ESS > 400). Support of topologies was evaluated in STACEY by constructing a tree of maximum reliability in TreeAnnotator (part of BEAST2 package) after rejection of a half of all estimated trees. Species delineation (based on the trees evaluated in STACEY) was carried out using a Java-application ‘speciesDA’ (http://www.indriid.com/software.html).

An input consensus multigene ultrametric tree was the same for “tr2” and STACEY. For this, we combined both sequences in the unified supermatrix via SequenceMatrix v.1.8 (Vaidya, Lohman & Meier, 2011), a nucleotide substitution model for each locus was defined in ModelFinder (for the entire 16S sequences and individually for each nucleotide position in the triplet for COI). We deliberately did not delete any sequences with incomplete and missing data, because this approach can greatly reduce the accuracy of the tree reconstruction (Molloy & Warnow, 2018). Further analysis was made in the same way as for phylogeny reconstruction, in BEAST2, but with 100M of MCMC generations and sampling every 100k tree. Due to a high uncertainty of the reconstructed tree, we used a final 50% annealing; in our subsequent analysis we used 500 trees from each run. However, the concatenation-based approach (Rokas et al., 2003) is reasonably criticized due to existence of a convergence between restored gene trees and the common species tree (Maddison & Wiens, 1997). To derive a species tree from these different gene trees, we used the multiple fusion technique implemented in ASTRAL-III v.5.6.3 (Zhang et al., 2018). No significant differences between the results of two analyses were found in Dendroscope (Huson & Scornavacca, 2012), so we used the BI tree for further conclusions.

In order to reduce the impact of a population structure to phylogenetic reconstructions, we previously divided the entire dataset into morphologically defined groups (excisa-like, exigua-like and nana-like) (Tables 1 and 2). Calculations of the nucleotide diversity indices (Nei & Kumar, 2000), demographic indicators (mismatch distribution and coalescence modeling for population growth and divergence) and the neutrality tests were performed in dnaSP v. 6.12 (Rozas et al., 2017). In order to check the neutrality of the loci and roughly describe possible demographic processes, we carried out the Fs test of neutrality (Fu, 1997) and R2 statistics (Ramos-Onsins & Rozas, 2002) as the best ways of such analysis (Ramírez-Soriano et al., 2008; Garrigan, Lewontin & Wakeley, 2010). The platform MEGA-X (Kumar et al., 2018) was used to calculate genetic distances. We selected “simple”p-distances as more preferable for DNA barcoding (Collins et al., 2012), as there is no significant difference between uncorrected and corrected substitution models in case of a sufficiently large dataset (Nei & Kumar, 2000).

Table 1 Genetic diversity of Alonella complexes.

Groups	N	G+C	n	S	h	Hd	Pi	k	Fs	R2	
16S (mitochondrion, rDNA)	
total 16S	83	0.342	409	184	53	0.985	0.182	70.3	1.039	0.189	
excisa complex	58	0.346	409	155	36	0.978	0.173	67.8	3.778	0.217	
exigua complex	17	0.320	409	52	12	0.941	0.042	16.8	0.497	0.158	
nana	8	0.341	390	125	6	0.893	0.156	61.1	5.132	0.221	
COI (mitochondrion, coding)	
total COI	78	0.374	626	202	40	0.974	0.166	83.4	5.128	0.204	
excisa complex	74	0.365	474	184	40	0.965	0.156	74.2	7.040	0.199	
exigua complex	3	0.373	626	86	3	1	0.037	48.8	5.279	0.357	
nana	2	0.377	626	0	1	–	–	–	–	–	
Notes.

N - number of sequences; G+ - guanine-cytosine content; n - total number of sites (excluding sites with gaps / missing data); S - number of segregating (polymorphic) sites; Hd - haplotype diversity; h- number of haplotypes; Pi - nucleotide diversity per site; k - average number of nucleotide differences; Fs - Fu’s neutrality statistic (Fu, 1997); R2- Ramos-Onsins and Rozas R2-statistic (Ramos-Onsins & Rozas, 2002).

Results

The 16S fragment was successfully amplified and sequenced from most studied individuals. A high rate of unsuccessfully PCRs for COI fragment (27%) is resulted presumably from the presence of sequence mutations at the primer binding sites, as even “universal primers” (Prosser, Martínez-Arce & Elías-Gutiérrez, 2013; Elías-Gutiérrez et al., 2018) did not work properly. The alignment contained 82 newly obtained 16S rRNA sequences (400 bp), 39 original (626 bp) and 39 previously obtained COI sequences deposited in NCBI Genbank or BOLD (Table S1).

Both loci were characterized by a relatively high haplotype and nucleotide diversity and a relatively low G+C portion in the coding COI locus, that, along with previous data (Kotov et al., 2016), may be characteristic of the chydorids in toto. Results of the neutrality tests for different loci for different groups of populations may indicate multidirectional demographic processes in different lineages and in different loci. Thus, the values of Fs>>0 at R2>0 were characteristic for both loci of the excisa-like taxa and may indicate a significant genetic differentiation within this group (with the possibility of splitting/mixing processes in the populations). The exigua-like group looks more homogeneous, and the high values of Fs and R2 for COI can be explained by an effect of the small sample size. However, these results demonstrated the need to study in detail the genetic structure of large groups of Alonella populations and to resolve the taxonomic uncertainty in these lines.

Table 2 Estimates of evolutionary divergence over sequence pairs between Alonella complexes.

We used uncorrected p-distance (Nei & Kumar, 2000). All ambiguous positions were removed for each sequence pair (pairwise deletion option). On this table COI are located above diagonal, 16S - below diagonal. In the line are within groups p-distance for 16S / COI respectively.

	outgroup	excisa	exigua	other	
outgroup	out	0.219	0.209	0.216	
excisa	0.216	0.16 / 0.15	0.202	0.203	
exigua	0.236	0.197	0.09 / 0.09	0.048	
nana	0.233	0.251	0.229	0.01 / 0.01	

Our original sequences together with the GenBank sequences could be assigned to 12 phylogenetically divergent clades, well supported statistically. We numbered all major clades by capital letters from “A” to “L”, the clades A–K are primarily defined based on the variation in the 16S tree (Fig. 2). The clade L is present in the COI tree only (Fig. 3) due to lack of 16S sequences from Mexican populations. In the COI tree, only 8 major clades were represented: B, C, D, E, G, H, J, L (Fig. 3), as we failed to obtain the sequences for clades A, F, I and K. In total, we differentiated a single major clade (A) for A. nana, two major clades (B–C) of the A. exigua complex and nine major clades (D–L) of the A. excisa complex (Figs. 2 and 3).

Figure 2 Maximum likelihood tree representing the diversity among phylogroups of Alonella based on 16S data.

The support values of individual nodes are based on bootstrap-test UFBoot2).

Figure 3 Maximum likelihood tree representing the diversity among phylogroups of Alonella based on COI data.

The support values of individual nodes are based on bootstrap-test UFBoot2.

Alonella nana (Fig. 4, upper panel, Table S1). Clade A was represented by two regional subclades: A1 was found in Europe and A2 was found in a single locality in North America.

Figure 4 Distribution of major Alonella clades (both original and sequences retrieved from NCBI GenBank).

The base map was from the open domain plain map available at https://marble.kde.org/.

A. exigua complex (Fig. 4, upper panel). Clade B was widely distributed through the northern Palaearctic; sequences from Siberia, Mongolia and European Russia form a subclade B1, while a single Central European sequence formed a separate subclade B2 (The subclade B1 was paraphyletic in the 16S tree) (Fig. 2). The clade C was found in several localities of northern North America forming a single subclade C1 (Newfoundland, Manitoba Ontario).

A. cf. excisa complex (Fig. 4, upper and bottom panels, Table S1). Clade D was present in the Northern Palaearctic from European Russia to Kamchatka, and also found in Arctic Canada (Fig. 4, upper panel); there were three subclades within the latter: D1 (northern portion of European Russia and Western Siberia), D2 (Manitoba) and D3 (Western Siberia, Eastern Siberia and Kamchatka) (the subclade D3 was paraphyletic in the 16S tree). Clade E was locally distributed in the southern portion of the Russian Far East and South Korea; all sequences formed a single subclade E1. Clade F (with a single subclade F1) was represented by a single sequence from Manitoba (Fig. 4, upper panel). Clade G (Fig. 4, bottom panel) was distributed in Siberia; it was represented by subclade G1 (Western and Eastern Siberia) and G2 (Western Siberia). Clade H (Fig. 4, bottom panel) was exclusively European, and was represented by subclade H1 (Central Europe and European Russia) (this subclade was paraphyletic in the 16S tree) and H2 (Central Europe only). Clade I was widely distributed in the eastern portion of Eastern Siberia, Russian Far East and Manitoba—it formed the subclades I1 (Eastern Siberia and Russian Far East) and I2 (Eastern Siberia and Manitoba) (the latter was paraphyletic in the 16S tree). Clade J was found only in the Ethiopian Bale Mountains, it contained two subclades (J1 and J2) both from this local area. Clade K (with single subclade K1) was found in a single locality in Ontario. Clade L (with single subclade L1) was found in the Yucatan Peninsula (tropical Mexico) (Fig. 4, bottom panel).

Genetic differentiation between the major clades was great (p-distance was 12.1–25.1% for COI, and 10.6–27.2 for 16S, Table S2) as compared to other invertebrates. Such a level of genetic differentiation corresponded to (at least) the species level, even if we applied the highest threshold values of such differences for the invertebrates (Hebert, Ratnasingham & De Waard, 2003; Hebert et al., 2004).

The tanglegrams for mitochondrial genes (Fig. S1 on-line), and species (Fig. S2 on-line) had similar topologies, including the terminal branches. There was strong agreement for the existence of the same major clades in both mitochondrial loci (Fig. S3 on-line). A few discrepancies in tree topologies were detected; however, it was clear that a reliable reconstruction of the tree branching at the high hierarchical level would benefit from a full representation of all clades. Such sampling could not be provided for objective reasons, i.e., part of the data was sourced from previously published studies. Delimitation on the entire general multi-locus tree via different methods, including those based on multi-species coalescence, was illustrated in Fig. S3 on-line. Major clades were recognized as separate units by all algorithms.

Discussion

Genetic basis for biodiversity understanding

Based on a logics of the “standard screening threshold of sequence difference (10 × average intraspecific difference”) (Hebert et al., 2004), we would have to conclude that each the excisa, exigua and nana is represented by a single polymorphic species. However, the levels of divergence in these complexes are significantly higher than it was previously found in most animals (Ratnasingham & Hebert, 2013; Meier et al., 2006; Huemer et al., 2014; Čandek & Kuntner, 2015). Therefore, an alternative and much more realistic explanation is a high cryptic variability within each studied complex. Possible morphological differences within the aforementioned species complexes must be studied in detail.

Different delimitation approaches result in different number of distinct units, which may possibly represent species (Fig. 5). The ABGD approach is known by its excessive splitting of the groups with high levels of polymorphism, and as a result, even “good” morphospecies could be easily split into several groups, as it was already shown for Daphnia magna (Bekker et al., 2018). Theoretically, these problems may have been resolved by using coalescent methods. However, there are obvious “excesses” of such packets as ’splits’ and ‘tr2’, working on the simplest algorithm without an opportunity to correct model parameters based on knowledge about the animal biology. As expected (Sukumaran & Knowles, 2017), GMYC models in case of Alonella tend to recognize some structured populations as real distinct species (Fig. 5). There are also well-known methodological problems concerning the GMYC (Maddison & Wiens, 1997; Powell, 2012; Reid & Carstens, 2012) and PTP (Zhang et al., 2013; Kapli et al., 2017) applications. Usually mPTP delimitation is more conservative, only the large groups of populations (Fig. 5) are recognized as species which allows to prevent an excessive splitting (Tang et al., 2014; Vitecek et al., 2017). But based on both the analysis of individual trees and species coalescence via several genes, we can state the presence of a complex species structure within the Alonella genus. Moreover, the main phylogenetic lineages are supported in all analyses.

Figure 5 Summary of results of molecular species delimitation via different methods.

The BI multi-locus tree is showed. Analyses referring to STACEY and “tr2” are based on multi-locus datasets; for further analytical details, see text. Coloration indicates group membership of specimens; absence of coloration indicates missing data. Node supports are UFboot2 (ML) and posterior probabilities (BI), in percent. Grey color marked absent sequences.

As it was shown above, the lack of data on a mitochondrial locus can be compensated via data on another locus, it allows us to carry out a reconstruction of the phylogenetic relationships based on the mitogenomes of Alonella, to identify main phylogenetic lines, potential OTUs, although they don’t have to be recognized as “biological species” (Blaxter et al., 2005).

Preliminary notes on biogeographic patterns in Alonella

Our initial study could not describe fully the biogeographic patterns and phylogeographic scenarios within Alonella (i.e., due to obvious sampling limitations), but some preliminary conclusions can be drawn. A separate issue is the possible effect of biological invasions on the formation of modern biogeographic patterns in some clades.

We can classify the patterns of major clades into six groups (Fig. 4):

(1) Trans-Beringian (“Holarctic”) (A, D, I);

(2) Palaearctic (B, G, H) - among which B and G are widely distributed, and H is exclusively European;

(3) Southern Far Eastern (E);

(4) Nearctic, with a pattern unknown to date due to a very limited set of samples from North America (C, F, K);

(5) Mexican Neotropical with unknown real range (L);

(6) Possible endemic Ethiopian (J).

These patterns may indicate a complicated history of dispersion and speciation for Alonella. But geographic patterns for the divergent and minor clades of Alonella are concordant to those from other cladoceran macro-taxa. A trans-Beringian distribution was observed in some clades of the Polyphemus pediculus group (Xu et al., 2009) and the Chydorus sphaericus group (Belyaeva & Taylor, 2009; Kotov et al., 2016). Clade B is widely distributed in the Northern Palaearctic (in our samples, from the Czech Republic to Yakutia Republic), while its sister clade C seems to be restricted to the North America, although presence of both clades in the Far East is also possible. The same situation is observed in the Moina macrocopa species group, where M. macrocopa s.str. is widely distributed in the Northern Palaearctic, while M. americana is restricted to the New World (Montoliu-Elena, Elías-Gutiérrez & Silva-Briano, 2019). The clades widely distributed through all the whole northern Palaearctic are also known for the Polyphemus pediculus group (Xu et al., 2009), Daphnia curvirostris group (Kotov & Taylor, 2019; Kotov et al., 2020), D. pulex group (Crease et al., 2012; Ballinger et al., 2013) and D. longispina complex (Yin et al., 2018; Zuykova et al., 2018). Exclusively Nearctic clades are found within many taxa (Bekker, Kotov & Taylor, 2012; Xu et al., 2011). Clade E is found in the southern Far East. According to our data, it is distributed from South Korea to Primorski Territory of Russia, but potentially, this taxon may have a wider distribution range as records of thermophilic Oriental taxa in the southern portion of the Russian Far East are not rare (Kotov, 2016). But, most probably, this clade belongs to an endemic Far Eastern faunistic complex (Kotov & Taylor, 2019; Kotov, 2016).

An example of the Alonella endemism is presumable a specific major clade J from Ethiopian high mountains. In Ethiopia, populations, belonging to the clade J, were detected from the same water bodies where another local endemic, Daphnia izpodvala, was found (Kotov & Taylor, 2010) and they are never found in the tropical lowlands. At the same time, reliable records of Alonella populations from other African countries are very limited. Such records are known from Chad (Rey & Saint-Jean, 1968), Fouta Djalon and adjacent mountain areas (Dumont, 1981), Cameroon rain forests (Chiambeng & Dumont, 2005), Rwenzori mountains (Van Damme & Eggermont, 2011). All these populations have not been studied via genetic methods yet.

Surprisingly, during our study we found some cases of trans-continental geographic ranges in Alonella (but only within the Holarctic). Thus, the European subclade A1 is a sister group to A2 from North America (USA, MA) (Fig. 4, upper panel; Fig. S2 on-line). Most likely explanation lies in some past dispersion scenario, with subsequent independent genetic evolution of these newly established populations. Such cases were previously demonstrated for other cladocerans (Taylor & Hebert, 1993; Marková et al., 2007; Millette et al., 2011).

Appearance of the lineages C and F in Canada could be explained as a result of a trans-Beringian transition (Fig. 4, upper panel). Such a transition possibly took place only around 20 thousand years ago, which corresponds well with existence of a massive land bridge between Eurasia and North America, Beringia. The level of genetic differences between them and their sister groups (lineages B and E, respectively) in Eurasia is comparable to that between A1–A2. The presence in Canada of the haplotype from the I2 subclade, close to the Yakutia-Primorsky haplotypes, could be also a consequence of recent anthropogenic introduction from Pacific Asia. Similar patterns are known for other freshwater microcrustaceans (Ishida & Taylor, 2007).

The phylogeographic situation is complicated in Canada (Fig. 4, upper and bottom panels; Fig. S2 on-line), where several sympatric main clades and/or subclades were found (D2, K1, F1, C1, I2). The K1 lineage is probably ancestral to the rest of the Alonella s.lat. taxa, and its status must be specially checked. The D2 clade is probably a North American phylogenetic lineage of the widespread circumpolar group D, such patterns are already found in Polyphemus pediculus (Xu et al., 2009). Probably, the subclade C1 and F1 are derived, respectively, from the Eurasian group of populations B and the Far Eastern group E. The comparable genetic distances between these North American and their ancestral groups may be a consequence of their appearance as a result of a trans-Beringian transition. Unfortunately, the Beringian zone is not sampled here, but Beringia apparently has an important role in the Alonella biogeographic patterns and needs to be specially studied in the future based on numerous samples.

Cryptic diversity of Alonella across the Northern Hemisphere and short comments on the inter-generic subdivision of Alonella

Our study confirms the opinion that the real diversity of the water fleas is several times higher than it is accepted now (Adamowicz & Purvis, 2005). This situation is usual for freshwater animals of different groups (Mills et al., 2017; Schwentner et al., 2020). We found several possible cryptic species within A. excisa and A. exigua species complex. To date, characters of the parthenogenetic females have a very limited value for the species discrimination within the A. excisa and A. exigua. Apparently, incorporation of males to morphological analysis may improve the situation, as it was already shown for some other chydorids (Belyaeva & Taylor, 2009; Kotov et al., 2016; Garibian et al., 2018), but, unfortunately, males only sporadically occur in the natural populations of Alonella, and, despite our significant efforts, we have no materials with males from some interesting localities, such as Ethiopia.

Before 2010, it was universally accepted that Alonella was a monophyletic genus, although the delineation between several chydorid genera (Alonella, Disparalona, Pleuroxus and Picripleuroxus) has been intuitive rather than based on accurate diagnostics (Neretina et al., 2018). Hudec (2010) subdivided the European taxa of the genus Alonella into two subgenera, Alonella s.str. and Nanalonella. The latter taxon has included a sole species, A. (N.) nana, with a globular shape of body, a single minute tooth on posteroventral portion of valve and a very short subquadrangular postabdomen. According to Hudec (2010), Alonella s.str. has included in Europe two species: A. excisa and A. exigua. Both morphospecies are characterized by a somewhat longer, oval body, a somewhat longer, angular postabdomen and posteroventral portion of valve with one or more denticles. Our data suggest that A. nana (clade A) is a sister group to A. exigua complex (clade B and C) (Fig. 2), in conflict with the subgeneric proposal by Hudec (2010). Variability in the number and shape of these denticles in some chydorids was previously discussed by many authors (Smirnov, 1996; Kotov, 2013; Neretina & Kotov, 2015; Neretina et al., 2018), and this feature seems dubious for a reliable discrimination of any subgenera. The same situation concerns the proportions of body and postabdomen. Another strong defect of such classification (Hudec, 2010) is that one ignores completely any Non-European Alonella taxa.

In fact, morphological differences between the best known Alonella species (Smirnov, 1996) are less expressed than those between Pleuroxus s.l. All attempts to subdivide the latter genus into several genera or subgenera by morphological criteria are controversial due to the mixing of morphological characters in the different taxa (Smirnov, 1996; Chiambeng & Dumont, 2004). The taxonomic challenges for Alonella and Pleuroxus must be resolved with a combination of morphological and genetic data (integrative approach), such studies are known for different microcrustacean groups (Karanovic & Cooper, 2012; Montoliu-Elena, Elías-Gutiérrez & Silva-Briano, 2019; Ni et al., 2019). Among the inter-generic subdivisions based on morphological characters carried out in the last two decades for any genera of the subfamily Chydorinae, only attempts to subdivide Disparalona s.l. may be considered successful due to the large number of reliable diagnostic features (Neretina et al., 2018; Neretina et al., 2019). In general, since the time of Smirnov (1996), morphological taxonomy of Chydorinae is poorly developed. For the latter, the morphological evidence is at its resolution limit, and such studies need to be coordinated with molecular studies.

Conclusions

Our study reveals a high cryptic diversity within the genus Alonella across the Northern Hemisphere. Some of detected main clades have wide ranges across the Old World (and even in the New World), others clades have more restricted ranges, or are likely endemics. Our results could be the basis for subsequent morphological study of Alonella, formal description of new taxa and subsequent biogeographical analyses. Thus, biogeographic study is possible for even the smallest of water fleas, as it was also demonstrated for other minute animals, like rotiferans (Cieplinski, Weisse & Obertegger, 2017; Mills et al., 2017) or ostracods (Hiruta et al., 2016). In this sense, “Little pigeons can carry great messages”.

Supplemental Information

Supplemental Information 1 Tanglegram for mitochondrial 16S (left) and COI (right) phylogenetic trees

Only unique sequences are represented.

Click here for additional data file.

Supplemental Information 2 Tanglegram for species trees reconstructed by merging in ASTRAL-III (left) and concatenation in BEAST2 (right) with subclades groups

Branches support for ASTRAL-III was bootstrap (100 replicas), for BEAST2 was posterior probabilities.

Click here for additional data file.

Supplemental Information 3 The individual ultrametric BI-trees for 16S (left) and COI (right) loci

Node supports are UFboot2 (ML) and posterior probabilities (BI).

Click here for additional data file.

Supplemental Information 4 Complete list of sequences obtained in the frame of this study and extracted from the GenBank and BOLD with information on specimen ID and locality provided for each individual

Clade designations correspond to those in other tables.

Click here for additional data file.

Supplemental Information 5 Genetic p.-distance between Alonella’s main clades

There are mitochondrial loci above diagonal –COI, below diagonal –16S in the table. In the line are within groups p-distance for 16S / COI, respectively.

Click here for additional data file.

We are much indebted to DJ Taylor for editing of earlier draft, EN Abramova, AN Babenko, EI Bekker, FE Fedosov, AS Golubtsov, FV Kazansky, BF Khassanov, NM Klimovskiy, NM Korovchinsky, NA Kuznetsova, OA Krylovich, ES Preobrazhenskaya, LV Razumovsky, VL Razumovsky, AB Savinetsky, LE Savinetskaya, DJ Taylor, AV Tchabovsky, EI Zuykova for ethanol-fixed samples with Alonella, HG Jeong for help during AAK’s sampling in South Korea, AA Darkov for help during AAN stay in Ethiopia. Special thanks to Prof. A Petrusek for organization of molecular analysis of some specimens in Charles University in Prague. Also, we thank SI Metelev and A.N. Nekrasov for their assistance to ANN during SEM works. All SEM works were carried out at the Joint Usage Center “Instrumental Methods in Ecology” (A.N. Severtsov Institute of Ecology and Evolution of Russian Academy of Sciences).

Additional Information and Declarations

Competing Interests

Author Contributions

Field Study Permissions

DNA Deposition

Data Availability

The authors declare there are no competing interests.

Anna N. Neretina conceived and designed the experiments, performed the experiments, prepared figures and/or tables, sEM studies, and approved the final draft.

Dmitry P. Karabanov analyzed the data, prepared figures and/or tables, and approved the final draft.

Veronika Sacherova conceived and designed the experiments, performed the experiments, authored or reviewed drafts of the paper, and approved the final draft.

Alexey A. Kotov analyzed the data, prepared figures and/or tables, authored or reviewed drafts of the paper, and approved the final draft.

The following information was supplied relating to field study approvals (i.e., approving body and any reference numbers):

Field collection in public property in Russia does not require permissions. Sampling in the state natural reserves of Russia was conducted with special verbal permission of their heads (O.P. Elizarova, Pinezhsky State Natural Reserve; T.I. Shpilenok, Kronotsky Biosphere Reserve; Y.P. Sushitsky, Khanka Nature Reserve). Sampling in Ethiopian was performed in the frame of the Joint Ethio-Russian Biological Expedition, curated by the Ministry of Environment, Forest and Climate Change of Ethiopia, and does not require special permission. Sampling in Mongolia was performed in the frame of the Joint Russian-Mongolian Complex Biological Expedition, curated by the Ministry of Nature, Environment and Tourism of Mongolia, and does not require especial permission. Samples in South Korea were collected in the frame of cooperation between A.A. Kotov and the National Institute of Biological Resources of Korea and does not require special permission. All the localities in Ethiopia, Mongolia and South Korea belong to public property. They are relatively large lakes, streams, affluents of rivers or roadside ponds.

The following information was supplied regarding the deposition of DNA sequences:

DNA described here are accessible at GenBank: MN608113- MN608151 (COI) and MN598677- MN598759 (16S).

The following information was supplied regarding data availability:

Raw data are available in the Supplementary Files.

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
