# Peer review of "Unexpected mitochondrial lineage diversity within the genus Alonella Sars, 1862 (Crustacea: Cladocera) across the Northern Hemisphere"

_PeerJ, doi:10.7717/peerj.10804_

## Round 0.1 · original submission · Minor Revisions

Thank you for resubmitting your manuscript to PeerJ. I have sent your paper to two expert referees for their consideration. I have now received their comments back and have read through your paper carefully myself. Enclosed please find the reviews of your manuscript.

The reviews are favourable and suggest that, subject to minor revisions, your paper could be suitable for publication. You are almost there, please consider these suggestions, and I look forward to receiving your revision.

Reviewer 1 ·

Basic reporting

The manuscript by Neretina et al. provides first larger-scale data on genetic diversity of the smallest water flea in the world, highlighting its substantial unrecognized diversity. The dataset is very valuable, the results are novel and to the body of evidence about the real diversity and biogeographic patterns in freshwater zooplankton and cladocerans in particular. The manuscript is certainly worth publishing, though I believe some revisions are either needed or at least recommendable before acceptance (see below, and the annotated PDF).

Experimental design

The study is based on material from many parts of the world available to the authors, the molecular analyses focus on genes for which some reference data are available, and they are rigorously analyzed, though I have a feeling that some analyses are in fact unnecessary.
In some sections, the manuscript is too detailed. This is particularly true for the Methods – some details about the molecular analyses are trivial and may be omitted (but elsewhere some extra information is needed), and the data analysis is described in extreme detail. The authors have used many different methods but in some cases I believe the additional analyses do not provide any added value.
In fact, I totally fail to understand why the “Population analysis” section is included – the dataset is not particularly suitable for population-level analyses or consideration of past demographic changes, these analyses are not necessary for delivering the message, nor are appropriately discussed. Furthermore, the authors may consider removing some species delimitation methods (in particular those that tend to result in over-splitting, as is discussed in the manuscript; sticking to methods that tend to be conservative might be sufficient for this particular story).
Presenting the results in the form of network seems also unnecessary, considering the high divergence of the individual clades (and thus uncertainty associated with exact network topology). The biogeographic patterns may be easily shown on the trees if the origin is highlighted by colours.
Overall, I believe the number of figures in the supplement may be reduced, some descriptive data in tables may not be necessary, and the Table 3 would rather fit to supplement.

Validity of the findings

The main conclusions are clear, and discussed appropriately. Sometimes, however, I feel the authors discuss even aspects or concepts that are not needed (e.g., focusing on an approach suggested in obsolete papers, or discussing in length the problems of subgeneric splitting without explaining why we need such splitting in the first place). Some of the aspects of biogeography or endemism seem to me overinterpreted, though, considering the limitations of the dataset. I point out to these specific issues in the annotated PDF.
I was not convinced by the level of splitting of the Alonella clades that are indicated in the trees and discussed in the main text. In some cases, the splitting to subclades does not seem to me as supported by the data on either gene (see, e.g., C1 vs C2, or D3 vs. the rest of the D clade). I do not think these subclades are in fact needed to describe or interpret the resulting patterns, so I urge the authors to very carefully consider when such subdivision is really warranted.
Discussion might also benefit from some restructuring. Especially at the beginning, it should be stronger, highlighting the main findings. The present version starts with explaining what pseudogenes are, which seems to me unnecessary.

Additional comments

I am providing rather detailed annotations to the main PDF (including comments to figures and tables), which I urge the authors to consider carefully. These deal with various aspects, both conceptual (requesting clarifications, improving visualization, commenting on particular conclusions) and minor (correcting inappropriate statements, rewording suggestions, etc.)
In general, despite specific comments and occasional criticisms, I like the story and look forward to seeing it published in the near future.

Annotated reviews are not available for download in order to protect the identity of reviewers who chose to remain anonymous.

Reviewer 2 ·

Basic reporting

This paper provides molecular evidence for cryptic diversity within three species of Alonella in the Holarctic. The conclusions are solid as far as they go, being founded on a quite comprehensive suite of phylogenetic analyses based on two mitochondrial gene regions. Although the geographic coverage is wide, the sampling is sparse for many regions (particularly North America). Alonella excisa is the most diverse among the three morphospecies studied, as would already be expected from public data in the BOLD database. The genetic distances found among clades strongly suggest that they are in fact good species, although further studies (including nuclear markers as well as morphology) are needed to corroborate this. The authors also document clear geographic structure in the distribution of clades, which again supports the distinctness of the various clades.
The analyses conducted are diverse and elaborate, following much the same approach used in a recent paper published by partly the same group (Sinev et al 2020, Zootaxa 4767 (1): 115–137). I get the impression that the elaborate analyses make something of an overkill on this limited dataset, but then I am not familiar with all the approaches applied. The results from different analyses are congruent and convincing enough. Nonetheless, I believe the general phylogenetic picture presented may have to be modified when sequences from more regions are added. It seems particularly important to include more samples from the Nearctic. This is further underlined by the author’s conclusion that clade K is basic to all other Alonella taxa. Still, the present paper is a significant step on the way to a more complete picture, and is therefore welcome.
I have no major objections to raise. I find the text unnecessarily long and wordy, but realize that both writing in English as well as tradition may contribute to this. In the following I point out a few issues that I recommend be addressed.

Methods: The heading “Population analysis” seems misleading, since no analysis of population structure is presented. Rather, the stated issue is on measures to avoid potential influence of population structure on the phylogenetic reconstructions. The various indices referred to here were calculated for clades, not for populations. I suggest simply to remove the heading.

Results: The authors should provide information on fragment lengths, alignment lengths and gaps in the alignments. In Table 1, the column called “n” provides # of sites without gaps, which is something else. I also wonder why the number of sites given for COI are unexpectedly low - the primer set they applied normally yield >650 bp, also in the three Alonella morphospecies. This may be due to poor quality of some sequences, but should in any case be commented on. I also miss information on checking for stop codons in the sequences.

Discussion: Some of the biogeographic/historical explanations suggested may be premature, but I have no objection to have them included. Comparisons with similar patterns in other species are certainly of interest.

Figure 4, legend: Information on which marker (16S) which the clades are based on should be included (at least some of them are only based on 16S).
Figure 5, legend: Statements about coloration are unclear. Two shades of grey are visible, I suggest “Dark grey” and “Light grey” to make clear the meaning.
Tables 1 and 2: Please include information of what the “other” groups are.

Details and minor issues
L. 126: manufacturer’s instruction – which manufacturer?
L. 158-159: Somewhat weird sentence (first part) – I suggest «The most appropriate substitution models and partitioning schemes were found via ModelFinder …»
L. 198: Instead of «similar what allows» I suggest «similar, which allowed»
L. 236: Write “according to Reid & Carstens”
L. 238 & 239: Substitute “which” for “what”
L- 248-252: This sentence is both complicated and incomplete, please rephrase
L: 266: Substitute “had” for “have”
L. 267: Delete “was”
L. 310: write “unsuccessful PCRs”, and delete “are”
L. 318-319: I agree that grouping the animals into morphotypes is reasonable, but the statement seems superfluous here. The reference to tables 1 and 2 appears illogical, since they present only DNA results. I believe what you mean is that the molecular results support the initial grouping based on morphotypes. If so, this should be stated clearly.
L. 321: The G+C content refers to the DNA sequence, not the translated amino acid chain. I suppose what you mean is the coding COI locus.
L. 366: From Table 3, I found the range of p-distances is 12.1-25.1 % for COI, and 10.6-27.2 for 16S. I suggest you specify these values separately for each locus in the text here.
L374: Write “the same major clades in both mitochondrial loci”
L. 389: Substitute “which” for “what”
L. 402: “relatively accurately” is an unfortunate expression. Relative to what? There is no mention of stop codons or pseudogenes in the Methods or Results sections, and such information should be provided. Phylogenetic reconstructions can be influenced by many other factors including geographic sampling, selection of genetic markers, etc. My guess is that this mitochondrial phylogeny will be modified in the future particularly by expanding the geographic coverage.
L: 423-424: The expression “just add water” is obscure and should be avoided. Be more specific in this criticism
L. 428: Last word should be “applications” (plural)
L. 458: replace “meanwhile” with “while”
L. 462-463: “Exclusively Nearctic taxa are found within many taxa” just sounds odd – maybe replace the first “taxa” with “clades”?
L. 469: I suggest “An example” instead of “The example”
L. 482: “TYA” = thousand years ago? Better spell it out

Experimental design

Research aims are clearly stated and appropriate. This work is empirical rather than experimental, and the authors have drawn on the material and data available to them. As stated earlier, the analyses are elaborate and include several quite recently developed tools, and the results are convincing.

Validity of the findings

As previously stated, the conclusions are convincing. Speculations on biogeographic history and possible dispersal routes are relevant and acceptable.

Additional comments

No additional comments

---

## Round 0.2 · Minor Revisions

Thank you for resubmitting your manuscript to PeerJ. I have sent your paper to the same referees which evaluated your work at the first review round. I have now received their comments back and have read through your paper carefully myself. Enclosed please find the reviews of your manuscript.

Both reviewers agree that the manuscript has been significantly improved, but still find some minor corrections or issues to be addressed prior to acceptance of your paper. You are almost there, since the PeerJ doesn't allow to make significant modifications of the text after the "Accept" decision, I therefore kindly ask you to revise your manuscript following the suggestions by the reviewers, after what your paper could be suitable for publication. I look forward to receiving your revision and wish you and your coauthors all the very best.
Stay safe,

Reviewer 1 ·

Basic reporting

The revised manuscript has improved in most aspects, and while I still agree with the other referees that some of the thorough analyses may be a bit of an "overkill", considering the very clear patterns observed, there are only a few minor issues that need to be clarified or corrected before acceptance.

Experimental design

No further comment.

Validity of the findings

Perfectly adequate.

Additional comments

I am generally happy with the way how the manuscript was improved by the authors. In the annotated file provided along with the review, I have included various corrections, suggestions for rephrasing (sometimes, I believe rephrasing is necessary to avoid confusion or ambiguities), and requests for clarification. These are in the form of either comments or tracked changes, and I believe the authors will have no problem with dealing with them.

In most cases, when the authors did not agree with my previous comments, I accept their decision. However, I believe the opening of the discussion ("Based on a logics of the "standard screening threshold of sequence difference (10× average intraspecific difference") (Hebert et al., 2004), we would have to conclude that each the excisa, exigua and nana is represented by a single polymorphic species. ") is based on an outdated concept, and thus not suitable. The assumption that interspecific divergence should be 10x the intraspecific one, is not warranted - I see no reason why to disprove all the outdated concepts that have emerged sometimes in the past. I strongly suggest the authors skip this bit - it is like a breaking into a widely opened door...

In addtition to the issues in the annotated manuscript, I would like to highlight the need for a careful check of the supplementary information.
In the Supplementary table 1, there are some lines where neither COI nor 16S accession number is provided (but apparenlty the sequences were available, as the clades and subclades are indicated). They need to be filled in. There are also two empty sheets ("list", labelled in Russion) that are not necessary.

In the Supplementary table 2, at least rough coordinates for the Danish site might be provided. I also suggest that the authors include columns for GenBank and/or BOLD acc. no., so the potential users of this information do not have to extract them manually from the code column.
In fact, I believe both of the above-mentioned supplementary tables may be easily merged into one, which would make the work with the data much easier.

The Supplementary Table 3 is in the Word format, spanning multiple pages, and difficult to read. It would be better to provide it in a spreadsheet form as well.

Annotated reviews are not available for download in order to protect the identity of reviewers who chose to remain anonymous.

Reviewer 2 ·

Basic reporting

I reviewed a previous version of this manuscript earlier. Many of the comments I made then were intended to aid the authors in improving the language. While their response letter states that they have made corrections accordingly, this is not always the case. So, I recommend the authors to look through the previous suggestions again.

I still find the long and complex treatment of delimitation of putative species an overkill. The methods, results and discussion related to species delimitation make up a large fraction of the paper. Since the authors seem to insist on keeping this elaborate element, at least they should provide some statement of their motivation for including so many approaches that yield similar results. One of their conclusions is that missing data on one mitochondrial locus can be remedied by data on another mitochondrial locus (lines 397-398). In my opinion, their data do not allow for this conclusion, which must be based on analyzing a complete dataset (without ‘gaps’) for comparison with the incomplete alternative. While the results on delimitation of MOTUs are quite reasonable, new data on nuclear DNA as well as morphological variation are necessary for a final evaluation of the status of the various clades identified. I'm sure the authors agree to this, and certainly hope for their follow-up along such lines.

In reviewing the previous version, I pointed out that there was no mention of how gaps in the alignments of sequences were treated. This omission has not been corrected in the present version, but such information is required. There are often no gaps in CO1 alignments of close relatives, but this cannot be said for 16S alignments.

Some details:
L 42: Daphnia O.F. Müller, 1776 should be 1785 (Müller 1776 wrote Daphne, not Daphnia)
L. 87-88: I don’t understand how the reference to the FADA list (Kotov et al. 2013) supports this statement
L. 304: Difficult sentence. You need at least to add “and” before “39 original”
L. 367: Replace FigurebS4 with Figure S3
L. 509: The reference to Cieplinski et al (2017) is irrelevant here, as they studied rotifers, not microcrustaceans
Figure 4, lower panel: The symbol for the J clade should be a circle, not a square
Figure 5. I’m still not happy with the legend referring to coloration – see my recommendation on the previous version. The inclusion of colored symbols on the tree branches is very helpful. However, the placement of the symbols for the G and H clades cause confusion and should be simplified. I suggest placing the G symbol closer to the G clade, and the upper H symbol at the root of the H1 clade.
Supplemental figures: I can find no legends to these in this revised version

Experimental design

No comments

Validity of the findings

No comments

---

## Round 0.3 · accepted · Accept

Thank you for taking the time to revise and resubmit your manuscript. I have now read through your paper as well as your letter in response to the reviews. I think that you have successfully addressed all of the concerns raised very well, and would like to accept your manuscript for publication in PeerJ. Congratulations!

Thank you for all the hard work you have put into this. Your paper makes a strong contribution to the literature and I look forward to seeing it published.

Wishing you all the best and good health for the coming 2021!

Sincerely, Nikolay